# Randomised controlled trial of exercise to prevent shoulder problems in women undergoing breast cancer treatment: study protocol for the prevention of shoulder problems trial (UK PROSPER)

Julie Bruce,[1] Esther Williamson,[2] Clare Lait,[3] Helen Richmond,[1] Lauren Betteley,[1] Ranjit Lall,[1] Stavros Petrou,[1] Sophie Rees,[1] Emma J Withers,[1] Sarah E Lamb,[1,2] Alastair M Thompson,[4,5] on behalf of the PROSPER Study Group

¹Division of Health Sciences, Warwick Clinical Trials Unit, University of Warwick, Coventry, UK
²Nuffield Department of Orthopaedics Rheumatology and Musculoskeletal Sciences, University of Oxford, Windmill Road, Oxford, UK
³Gloucestershire Care Services NHS Trust, Gloucester, UK
⁴Department of Breast Surgical Oncology, University of Texas MD Anderson Cancer Center, Houston, Texas, USA
⁵Department of Translational Molecular Pathology, University of Texas MD Anderson Cancer Center, Houston, Texas, USA

**Correspondence to**
Dr Julie Bruce;
julie.bruce@warwick.ac.uk

## ABSTRACT

Musculoskeletal shoulder problems are common after breast cancer treatment. Early postoperative exercises targeting the upper limb may improve shoulder function. This protocol describes a National Institute for Health Research-funded randomised controlled trial (RCT) to evaluate the clinical and cost-effectiveness of an early supervised structured exercise programme compared with usual care, for women at high risk of developing shoulder problems after breast cancer surgery.

**Methods** This pragmatic two-armed, multicentre RCT is underway within secondary care in the UK. PRevention Of Shoulder ProblEms tRial (PROSPER) aims to recruit 350 women from approximately 15 UK centres with follow-up at 6 weeks, 6 and 12 months after randomisation. Recruitment processes and intervention development were optimised through qualitative research during a 6-month internal pilot phase. Participants are randomised to the PROSPER intervention or best practice usual care only. The PROSPER intervention is delivered by physiotherapists and incorporates three main components: shoulder-specific exercises targeting range of movement and strength; general physical activity and behavioural strategies to encourage adherence and support exercise behaviour. The primary outcome is upper arm function assessed using the Disabilities of the Arm, Shoulder and Hand (DASH) questionnaire at 12 months postrandomisation. Secondary outcomes include DASH subscales, acute and chronic pain, complications, health-related quality of life and healthcare resource use. We will interview a subsample of 20 participants to explore their experiences of the trial interventions.

**Discussion** The PROSPER study is the first multicentre UK clinical trial to investigate the clinical and cost-effectiveness of supported exercise in the prevention of shoulder problems in high-risk women undergoing breast cancer surgery. The findings will inform future clinical practice and provide valuable insight into the role of physiotherapy-supported exercise in breast cancer rehabilitation.

**Protocol version** Version 2.1; dated 11 January 2017
**Trial registration number** ISRCTN35358984; Pre-results.

### Strengths and limitations of this study

► A large pragmatic study delivering a complex intervention to prevent postoperative health problems in patients with newly diagnosed cancer within secondary care.
► A strength of the evaluation is the mixed methods approach incorporating embedded qualitative research and economic analysis.
► Recruited participants undergo multiple cancer treatments, thus experience a complicated postoperative recovery pathway.

## BACKGROUND

Breast cancer is the most common cancer in women in the UK with a 30% increase in incidence since the early 1970s.[1] Due to advances in screening and treatment, the survival rate has progressively risen and now two-thirds of women will survive for 20 years beyond their breast cancer diagnosis.[1] The mainstay treatment is surgery to the breast and axilla, supplemented with chemotherapy, radiotherapy (RT) and endocrine therapy, depending on tumour stage and other clinical criteria. As a consequence of these treatments, upper limb problems such as decreased shoulder range of movement (ROM), impaired strength, chronic pain, sensory disturbances and lymphoedema are common adverse treatment effects.[2,3] Studies suggest that up to 67% of women have arm or shoulder symptoms up to 3 years after treatment.[4] Persistent upper limb dysfunction and pain are debilitating, have a negative impact on sleep, quality of life, physical functioning and emotional well-being. Given successes in increasing survival, it is timely to identify

strategies to improve the health-related quality of life of women after breast cancer treatment.

Several risk factors for shoulder and upper body problems after breast cancer treatment have been identified, including treatment-related factors, such as type of axillary surgery and RT, and patient factors such as age, body mass index (BMI) and pre-existing shoulder problems.[4 5] Women undergoing mastectomy compared with breast conserving surgery are at greater risk of postoperative shoulder restrictions (OR 5.7, 95% CI 1.03 to 31.2).[4] Additionally, those undergoing axillary lymph node clearance (ANC) are at greater risk of postoperative arm complaints compared with those having sentinel lymph node biopsy (OR 9.8; 95% CI 3.5 to 27.5).[4–6] RT, particularly to the axilla or chest wall, increases the odds of shoulder restriction (pooled OR 1.7, 95% CI 1.0 to 2.9) and lymphoedema (pooled OR 1.5, 95% CI 1.2 to 1.8) compared with women treated without adjuvant RT.[4] Women reporting problems in the upper body and shoulder region before surgery are also at increased risk of chronic postoperative pain.[5 7] BMI at time of surgery has been shown to have an independent negative effect on shoulder external rotation up to 7 years after breast cancer treatment, and increased BMI is a risk factor for chronic postoperative pain and arm lymphoedema.[7 8] It is important that the UK National Health Service (NHS) provides optimal care for these women at high risk of developing shoulder problems to ensure recovery and return to usual activities after cancer treatment.

A Cochrane review identified 24 studies (2132 participants) investigating exercise following breast cancer surgery.[9] Six studies (n=354), conducted outside of the UK, found that structured postoperative exercise significantly improved shoulder ROM in the short term and long term when compared with usual care.[9] Ten studies (n=1304) have evaluated timing of exercise delivery; programmes initiated immediately postoperatively (1–3 days) versus delayed exercise suggest that early postoperative exercise does significantly improve long-term shoulder ROM. However, some studies reported an increased risk of wound-related complications with early exercise, such as seroma and surgical site infection.[9] The largest UK trial to date (n=116 patients), published after the Cochrane review, found that participants were less likely to develop lymphoedema when exercises were limited to 90° of shoulder elevation during the first postoperative week compared with those performing unrestricted exercises.[10] These previous trials investigating the efficacy of exercise following breast cancer surgery have been criticised for being of poor methodological quality and for omitting important patient-reported outcomes such as function and health-related quality of life.[9] Furthermore, there is ongoing uncertainty around the optimal type, dose and timing of exercise after breast cancer treatment. Moreover, none of the trials conducted to date has investigated the cost-effectiveness of structured exercise programmes after breast cancer treatment.

## Rationale for a trial

To date, no large-scale, high-quality, multicentre randomised controlled trial (RCT) investigating the clinical effectiveness of a structured physiotherapy intervention compared with usual care for women undergoing breast cancer surgery has been conducted. Given the lack of knowledge regarding the intensity and duration of exercise interventions after breast cancer surgery, this trial will provide evidence on whether a rigorously designed physiotherapy-led intervention, incorporating behaviour change theory, improves postoperative function and related outcomes, and whether this is cost-effective to deliver in the NHS setting.

## METHODS
### Aim

The overall aim of PRevention Of Shoulder ProblEms tRial (PROSPER) is to investigate the clinical and cost-effectiveness of an early supervised exercise programme compared with best practice usual care for women at high risk of shoulder problems after treatment for breast cancer, on outcomes of upper arm function, complications and quality of life. Specific trial objectives are:

1. To develop and refine a complex intervention of physiotherapy-led exercise for women at risk of developing musculoskeletal problems after breast cancer treatment;
2. To assess the acceptability of the structured exercise programme and outcome measures, optimise participant recruitment and refine trial processes during a 6-month internal pilot phase at three clinical centres;
3. Use findings from the internal pilot phase to undertake a definitive full RCT in approximately 15 UK NHS breast cancer centres.

This protocol follows guidance from the Standard Protocol Items: Recommendations for Interventional Trials (SPIRIT).[11] Core trial information is presented in table 1. Figure 1 as per SPIRIT guidance, details the schedule of enrolment, interventions and assessment.

## Trial design and setting

A multicentre, pragmatic, parallel, two-arm RCT with an internal pilot study and embedded economic evaluation and qualitative studies. The trial framework is superiority rather than equivalence or exploratory. The trial is currently open and recruiting from 17 NHS tertiary breast cancer centres across England. Participants are randomised in a 1:1 ratio between intervention and control arms.

## Patient and public involvement in trial design

Four female patient and public involvement (PPI) representatives, all of whom were treated for breast cancer,

| Table 1 | WHO trial registration data set |
|---|---|
| **Data category** | **Information** |
| Primary registry and trial identifying number | ISRCTN35358984 |
| Date of registration in primary registry | Project number 13/84/10 |
| Secondary identifying numbers | Health Technology Assessment (HTA) |
| Source of monetary or material support | National Institute for Health Research, HTA |
| Joint sponsor | University of Warwick/University Hospitals Coventry and Warwickshire NHS Trust |
| Contact for public queries | prosper@warwick.ac.uk |
| Contact for scientific queries | Professor Julie Bruce, Warwick Clinical Trials Unit, University of Warwick |
| Public title | Exercise to prevent shoulder problems in patients undergoing breast cancer treatment |
| Scientific title | The PRevention Of Shoulder ProblEms tRial: a randomised controlled clinical trial comparing physiotherapy-led exercise vs usual care in women at high risk of shoulder problems after breast cancer surgery |
| Countries of recruitment | UK |
| Health condition or problem studied | Breast cancer |
| Interventions | Advice only: breast cancer care leaflets<br>Comparator: physiotherapy-led structured exercise programme incorporating behavioural strategies |
| Key inclusion and exclusion criteria | Age: 18 years or over, no upper age restriction<br>Sex: female<br>Inclusion: confirmed invasive/non-invasive primary breast cancer schedule for surgical excision, at high risk of shoulder problems as defined by criteria given in table 2<br>Exclusion: males, and women with exclusion criteria as described in table 2. |
| Study type | Interventional<br>Allocation: randomised; individual assignment<br>Primary purpose: prevention<br>Phase III |
| Date of first enrolment | January 2016 |
| Target sample size | 350 |

Continued

| Table 1 | Continued |
|---|---|
| **Data category** | **Information** |
| Recruitment status | Recruiting to July 2017 |
| Primary outcome | Arm, shoulder and hand function as measured using the Disabilities of the Arm, Shoulder and Hand (DASH) questionnaire at 12 months |
| Key secondary outcomes | DASH subscales, pain (acute, chronic, neuropathic), health-related quality of life, surgical site infection, lymphoedema and other complications, healthcare resource use. Exercise/activity data to inform adherence to interventions |

were consulted during the initial grant preparation, intervention development and trial set up. Our PPI representatives contributed to the design of the intervention and advised on recruitment-related issues; they provided valuable insight into the worries and concerns experienced during cancer treatment.

### Eligibility criteria

Women are eligible to participate in PROSPER if they are: diagnosed with histologically confirmed invasive or non-invasive primary breast cancer scheduled for surgical excision; aged 18 years or over; can comply with the protocol; willing to provide written informed consent and considered as being at high risk of developing postoperative shoulder problems (table 2). This is a pragmatic trial and it is important that inclusion criteria reflect contemporary clinical practice. Therefore, women are also eligible where a later decision is made for postoperative RT to the axilla and/or supraclavicular region, thus changing their risk status from low to high. 'Late entry' women are eligible for the trial if the decision for postoperative RT is made within 6 weeks of surgery. Women who have had previous breast surgery (such as excision of a benign tumour or breast cyst) and those women who have had previous contralateral (opposite side) mastectomy, are eligible for invitation providing they fulfil high-risk criteria for shoulder problems. Women having immediate reconstruction or bilateral breast surgery are ineligible as the usual NHS postoperative care pathway often includes routine postoperative physiotherapy. Exclusion criteria are presented in table 2.

### Participant screening, recruitment and consent

Participants are screened and identified from multidisciplinary team meetings and preoperative breast/oncology clinic lists in secondary care. The initial screening process is undertaken by a member of the clinical team, research nurse or trained designee. Potentially eligible patients are approached by clinical or research staff and are given a patient information sheet with further explanation of the trial. Figure 2 summarises participant flow.

| | Study period | | | | |
|---|---|---|---|---|---|
| | Enrolment | Allocation | Post-allocation | | |
| Time point | -$t_1$ | 0 | $t_1$ 6 weeks | $t_2$ 6 months | $t_3$ 12 months |
| Enrolment: | | | | | |
| Eligibility screen | √ | | | | |
| Informed consent | √ | | | | |
| Randomisation | 350 | √ | | | |
| Interventions | | | | | |
| Usual Care (UC) | 175 | All participants | ←——→ | | |
| UC + PROSPER intervention | 175 | | ←————————————→ | | |

**Figure 1** Study outcome measures and assessment time points. PROSPER, PRevention Of Shoulder ProblEms tRial.

## Allocation sequence generation and randomisation

Randomisation is based on a computer-generated algorithm held and controlled centrally by the Warwick Clinical Trials Unit (WCTU) programming team, independent from the PROSPER team. The WCTU telephone randomisation service is used whereby

**Table 2** Trial inclusion and exclusion criteria

| Inclusion criteria | Exclusion criteria |
|---|---|
| Age ≥18 years | Males |
| Histologically confirmed invasive or non-invasive primary breast cancer scheduled for surgical excision of breast cancer | Women having immediate reconstructive surgery |
| Considered high risk of developing shoulder problems after surgery defined by one or more of the following: ► Planned axillary node clearance ► Planned radiotherapy to the axilla and/or supraclavicular* ► Existing shoulder problems (based on PROSPER screening criteria) | Women having SLNB, with or without breast surgery, unless they fulfil other high-risk criteria |
| Obesity defined as body mass index >30 | Women having bilateral breast surgery |
| Any subsequent axillary surgery related to primary surgery, eg, axillary lymph node clearance conducted after SLNB | Evidence of metastatic disease at time of recruitment |
| Able to provide written informed consent | |
| Willing and able to comply with the protocol | |

*Includes women informed of need for radiotherapy to the axilla and/or supraclavicular within 6 weeks of surgery, thus potential late entry to the trial is allowed in this setting.
PROSPER, PRevention Of Shoulder ProblEms tRial;
SLNB, sentinel lymph node biopsy.

randomisation occurs after eligibility and informed consent has been obtained. Concealment of allocation is maintained. An automated confirmation email of intervention allocation is generated to the study team. Randomisation is stratified by the following variables: (i) first versus repeat surgery; (ii) centre and (iii) whether informed of the need for RT within 6 weeks of surgery. The first variable adjusts for the requirement for any additional surgery which may change risk status from low to high (eg, second procedure ANC or re-excision of surgical margins). The second stratification variable ensures balanced allocation across each recruitment site. The third variable accounts for late entry to the trial, thus relates to the timing of intervention delivery and whether participants are randomised preoperatively (up to the day of surgery) or within the first 6 weeks postoperatively. Due to the nature of the study intervention, it is not possible to blind participants or treating physiotherapists to treatment allocation. However, receipt and handling of outcome data collection is blinded, thus data entry of returned postal questionnaires, data cleaning and interim statistical analyses are conducted without knowledge of treatment allocation (blinded).

## INTERVENTIONS
### Control arm: usual care

All participants allocated to the usual care arm receive best practice usual care in the form of written leaflets containing information about exercises, recovery after surgery and treatments for breast cancer. During the pilot phase, different exercise information leaflets were reviewed and considered; we also consulted best practice guidance for written patient information materials.[12] The most commonly used information leaflets were 'Exercises after Breast Cancer Surgery (BCC6)' and 'Your Operation and Recovery (BCC151)' published by Breast Cancer Care (BCC).[13] The BCC leaflets were selected because of content, style and clarity of presentation of information.

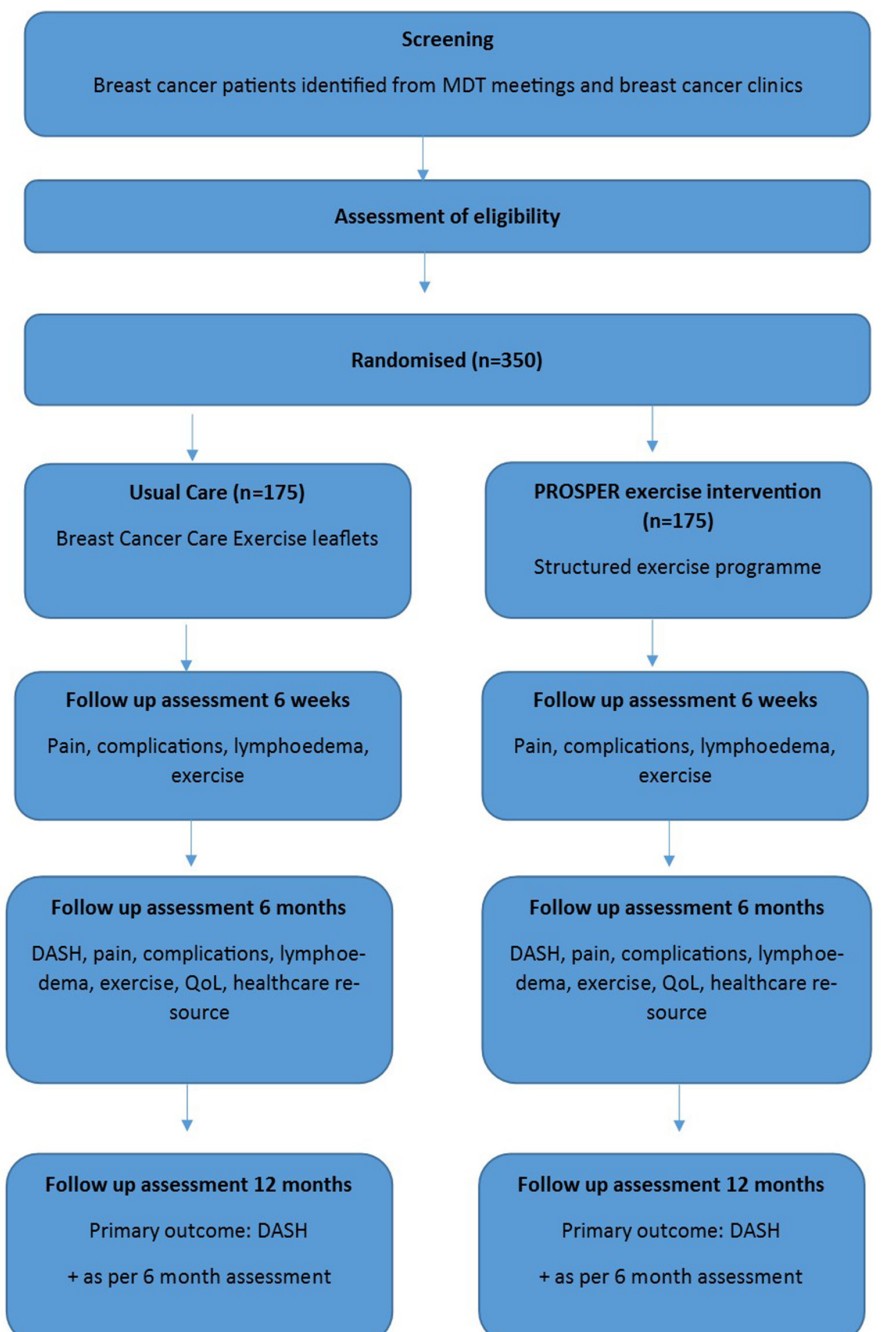

**Figure 2** Trial flow diagram. DASH, Disabilities of the Arm, Shoulder and Hand; MDT, multidisciplinary team; PROSPER, PRevention Of Shoulder ProblEms tRial; QoL, quality of life.

These two information leaflets were given to all patients before surgery by breast care nurses, or other healthcare professionals, depending on local practice.

### Intervention arm: PROSPER exercise programme

Participants randomised to the active intervention receive usual care leaflets in addition to the PROSPER intervention: a structured individualised exercise programme, comprising a minimum of three face-to-face and maximum of six sessions or contacts with a physiotherapist. As per Medical Research Council and TiDieR guidance, a more detailed description of the intervention development and final content has been described

separately (submitted for publication). We selected exercises and components based on systematic reviews and clinical guidelines. A Cochrane review investigated the effectiveness of exercise interventions in preventing, minimising or improving upper-limb dysfunction due to breast cancer treatment.[9] This review included 24 trials and classified exercise type as active, active-assisted, passive ROM, manual stretching, active stretching and resistance exercises. We considered these components in relation to evidence of effectiveness on shoulder ROM and strength. This process was also augmented by eliciting opinions from clinical experts in the field of cancer rehabilitation and health psychology. The final PROSPER programme

comprises specific exercises targeting shoulder range of motion and upper arm muscle strength, general physical activity and behavioural adherence strategies.

## Overview of exercise intervention

The intervention is predominantly delivered in physiotherapy outpatient departments. The first physiotherapy session is arranged 7–10 days after surgery, for assessment of shoulder ROM, postoperative pain, function, arm swelling, patients' goals and assessment of confidence to carry out prescribed exercises. Participants are prescribed an individually tailored home exercise programme and provided with guidance on rehabilitation, management of postoperative complications and returning to general physical activity and/or work. The intervention targets three movement directions using a combination of active-assisted ROM, active ROM and stretches: shoulder flexion (forward), shoulder abduction (side) and abduction with external rotation (open chest). The second appointment is between 4–6 weeks postoperatively to review progress and prescribe shoulder strengthening exercises. The programme is progressed by increasing exercise repetitions, sets and resistance. The third appointment is recommended for between 12 and 16 weeks postoperatively, for further progression to facilitate return to work, sport and hobbies. For women with later entry on the basis of postoperative RT, these timings will be slightly delayed, but the prescribed exercise programme should commence at the earliest opportunity, thus within 6 weeks of surgery.

As per development work with patient representatives, and to reflect the pragmatic trial design, three additional physiotherapy consultations are available on request. The timing and delivery of additional appointments, either via telephone or face-to-face, are flexible to account for ongoing treatment, physiotherapist judgement and patient preference. Ideally, the intervention will be completed within the first 6 months following surgery, but women can contact their physiotherapist for up to 12 months after randomisation. Thus, any late treatment-related shoulder problems will be dealt with by the trial physiotherapist. Number and method of physiotherapy contacts will be closely monitored during the trial.

## OUTCOMES

Figure 1 and table 3 present the study outcome measures and standardised assessment scales by assessment time point. Questionnaires are completed at baseline on recruitment, then at 6 weeks, 6 and 12 months after randomisation by post. The primary outcome is upper limb function at 12 months measured using the Disabilities of Arm, Shoulder and Hand (DASH) questionnaire.[14] We considered other patient-reported outcome measures, including shoulder-specific scales, however selected the DASH because it captures symptoms and function of the upper limb rather than the shoulder joint per se. There is good evidence to suggest that women experience a variety of difficulties and restrictions after breast cancer treatment, affecting the hand, arm and shoulder. Functional impairment to the arm can affect performance of simple daily activities, including writing, opening or closing jars, lifting and/or holding shopping bags.

The DASH is a 30-item patient-reported outcome measure designed to capture difficulty in performing various upper arm activities.[14 15] A single DASH score is generated, although psychometric assessment using discriminant content validation analysis has shown that the scale can be used to produce three health outcome subscores for impairment, activity limitation and participation restriction, as per the WHO International Classification of Functioning Disability and Health taxonomy.[16]

**Table 3** Outcome assessment

| Outcome | Domain | Scale/measure | $T_0$ baseline | $t_1$ 6 weeks | $t_2$ 6 months | $t_3$ 12 months |
|---|---|---|---|---|---|---|
| Primary | Function | DASH | | | | √ |
| Secondary | Function | DASH subscales | √ | | √ | √ |
| | Acute and chronic pain | FACT-B4; NRS | √ | √ | √ | √ |
| | Neuropathic pain | DN4 | √ | √ | √ | √ |
| | Complications | SSI+self-report | | √ | √ | √ |
| | Lymphoedema | Self-report | √ | √ | √ | √ |
| | Health-related QoL | SF12/EQ-5D-5L | √ | | √ | √ |
| | Resource use | Self-report | | | √ | √ |
| | General activity and exercise | PASE items | √ | √ | √ | √ |

DASH, Disabilities of Arm, Shoulder and Hand questionnaire; DN4, Doleur Neuropathique; EQ-5D-5L, Euroqol; FACT-B4, Functional Assessment of Cancer Therapy-Breast; NRS, numerical rating scale; PASE, Physical Activity Scale for the Elderly; QoL, quality of life; SF12, Short Form-12; SSI, surgical site infection; t, time point.

## Secondary outcomes

Secondary outcomes include health-related quality of life (EuroQol EQ-5D-5L and Short-Form-12), DASH subscores and surgical adverse events including pain (acute, chronic, neuropathic pain) surgical site infection and lymphoedema as per table 3. A numerical rating scale 0–10 and Doleur Neuropathique Questionnaire are used to collect pain intensity and pain character. The Functional Assessment of Cancer Therapy-Breast subscale captures arm tenderness, numbness, painful movement and stiffness. We added items to capture arm heaviness and swelling as self-report indicators of lymphoedema. Data on exercise/mobility are collected to allow comparisons in physical activity (selected items from the Physical Activity Scale for the Elderly (PASE)). The PASE was designed for use with older adults has been validated for use in clinical trials recruiting patients aged 55 years and older.[17] Healthcare resource use is recorded for economic analyses.

## SAMPLE SIZE

PROSPER aims to recruit 350 patients, allocated in a 1:1 ratio. The sample size calculation is based on a Dutch trial of 30 women with breast cancer, randomised to physiotherapy over a 3-month period, reporting a between-group difference of 7 points on the DASH at 6 months.[18] At 80% power and P<0.05, this yields a target of 242 participants in total. Accounting for therapist effects, an intracluster coefficient (ICC) of 0.01 (yielding a design effect of 1.05), gives a target of 256 participants. The ICC estimate is based on our previous experience of exercise interventions in a range of musculoskeletal trials. We anticipate loss to follow-up of <10% based on our previous clinical trials however, have inflated this to 25% to cover the possibility that numbers lost to follow-up are greater than anticipated, for example, due to ongoing cancer treatment.

The study is powered to detect a 7-point difference on the DASH. Studies of rheumatological and orthopaedic populations have suggested that the minimally clinically important difference for the DASH is 10, and that the between-group difference for trials should be set at 10.[19] However, this fails to account for many of the eventualities that occur in pragmatic trials, notably that there is not a 'no treatment' control arm, and therefore that some of the control group may be exposed by serendipity to an intervention of similar intensity, particularly in a high-risk population.

## Internal pilot study

A 6-month internal pilot phase was conducted at three breast cancer units (Coventry, Oxford and Wolverhampton) to evaluate processes for patient identification, eligibility and refinement of recruitment estimates. The intended sample size for the internal pilot study was 30 participants, approximately 10% of the full sample. Acceptability of the PROSPER intervention was explored through qualitative research involving audio-recorded individual interviews with seven participants. Changes were made to patient-facing materials and to exercise intervention materials. Easy-to-use pocket-sized laminated cards with details of inclusion/exclusion criteria and shoulder screening criteria were produced for recruitment staff. Additional telephone or face-to-face appointments were added to the exercise intervention to allow for flexibility during ongoing cancer treatment. Data from the pilot phase helped to refine recruitment and trial processes. Patients recruited to the pilot phase continue with the follow-up schedule and will be retained in the full trial analysis. The pilot study was completed as planned and the funder approved progression to full trial.

## Data analysis

Statistical analysis will be intention-to-treat and will comply with the Consolidated Standards of Reporting Trials (CONSORT) guidelines. The primary outcome data will be summarised using mean, SD, median and range values. The clustering effect will be assessed prior to analysis of the data. In the presence of a clustering effect, the primary outcome will be analysed using multilevel linear regression models. If there is negligible clustering effect, it will be analysed using ordinary linear regression models. In each case, the mean change from baseline (to 6 and 12 months) will be summarised for each of the treatment arms and differences between the interventions using unadjusted and adjusted (for age, type of surgery and RT) estimates. These mean changes and their 95% CIs will be plotted graphically so that change can be assessed over the course of the study. Continuous secondary outcomes will be assessed in a similar way to the primary outcome. Categorical data will be analysed using random effect/ordinary logistic models, depending on the presence of a clustering effect.

A DASH score cannot be computed if there are more than three missing items. As a sensitivity analysis, the impact of missing data will be assessed using multiple imputation. The impact of non-compliance with the intervention will be examined using the complier average causal effect (CACE) analysis.[20 21] We have reviewed definitions of compliance for CACE analyses used in other therapy trials.[22 23] Complete compliance with the PROSPER intervention is defined as having three or more contacts with the PROSPER therapist; an additional analysis will be undertaken to explore partial compliance, defined as less than three sessions. Analyses and template tables will be reported in a detailed statistical analysis plan for review and approval by the Data Monitoring Committee (DMC), prior to final statistical analysis of the data. Planned sensitivity analyses include: a) the impact of low/high recruitment centres on clustering effect and b) assessment of differences between date of randomisation and date of surgery across groups, as surgical trials vary in relation to timing of follow-up.

## Economic evaluation

The primary economic evaluation will be conducted from the NHS and personal social services perspective[24]

using the intention-to-treat approach.[25] Data will be collected on the health and social service resources used in the treatment of each trial participant from randomisation to 12 months postrandomisation. Primary research methods will be used to estimate the costs of delivering the physiotherapy-led exercise programme, including development and training of accredited providers, the cost of delivering the individual sessions and participant monitoring activities. Broader resource utilisation will be captured through three main sources: (i) clinical data extraction forms; (ii) patient postal questionnaires at 6 and 12 months postrandomisation; and if feasible within the trial timeline, (iii) routine health data sources from NHS Digital. Current UK unit costs will be applied to each resource item to estimate costs in each trial arm. Health-related quality of life will be measured at baseline and at 6 and 12 months postrandomisation using the generic EQ-5D-5L and SF-12 measures; national tariff sets will be used to generate quality-adjusted life-years (QALYs).[26–30]

An incremental cost-effectiveness analysis, expressed in terms of incremental cost per QALY gained, will be performed. Detailed methods of analysis will be prespecified within a health economics analysis plan approved by the trial team prior to analysis to ensure appropriate methods are used. Results will be presented using incremental cost-effectiveness ratios (ICERs) and cost-effectiveness acceptability curves generated via the net-benefit framework. A series of sensitivity analyses will be undertaken to explore the implications of uncertainty on the ICERs and to consider the broader issue of the generalisability of the study results. Due to the known limitations of within-trial economic evaluations,[31] a decision-analytical model may be constructed to examine the long-term costs and outcomes beyond the end of the trial. Costs and outcomes beyond the first year will be discounted to present values[24] and probabilistic sensitivity analyses will be undertaken to explore the impact of uncertainty on the ICERs.

## Qualitative substudy

An embedded qualitative study will be undertaken to gain insight into the experiences of women participating in trial interventions. We will explore the acceptability of the exercise programme and compare and contrast experiences with women allocated to the control intervention.

## Design of substudy

In-depth, semi-structured interviews will be conducted and audio-recorded. Interview topic guides will be used to ensure similar areas are covered in each interview. Participants consenting to the main trial are asked to indicate willingness to take part in a future interview to explore postoperative experiences. A total of 20 interviews are planned, with 10 women from each intervention arm. Purposive sampling will be used, striving for a mix of geographical location, age, employment status, socioeconomic background and ethnicity.

## Analysis

Interviews will be recorded, transcribed and analysed using a Framework Approach. A thematic framework will be developed using predetermined themes plus new themes raised by participants. The framework will be applied to the interview text and coded data will be arranged on a chart according to each theme identified. Themes will be examined with a view to providing explanations of the participants' experiences and understandings.

## Data security and management

Participant data are stored on a secure database in accordance with the Data Protection Act (1998). A unique trial identification number is used on all participant communication. Clinical and patient forms are being checked for completeness and congruity before data entry onto the PROSPER database. Data will undergo additional checks to ensure consistency between data submitted and original paper forms. Trial documentation and data will be archived for at least 10 years after completion of the trial in accordance with WCTU standard operating procedures.

## Trial monitoring

The Trial Management Group will oversee all aspects of design, delivery, quality assurance and data analysis. A Trial Steering Committee (TSC), with independent Chairperson, will monitor the trial at least once per year. An independent DMC will review trial progress, recruitment, protocol compliance and interim analysis of outcomes, annually or more frequently as requested. Recruitment data from the internal pilot study were reviewed by independent committees and by the funder to approve the launch of the main trial.

## Adverse event management

A safety reporting protocol has been developed for related and unexpected serious adverse events and directly attributable adverse events (AEs). An AE is defined as any untoward medical occurrence in a subject which does not necessarily have a causal relationship with the intervention. Any AE that occurs while undertaking PROSPER exercises, either during an appointment, or while exercising unsupervised at home, require reporting to the trial team. The trial Chief Investigator, with input from the WCTU Quality Assurance team, determine whether AEs require reporting to the trial sponsor, DMC and Ethics Committee, in accordance with the full safety reporting protocol.

## Dissemination policy

The study team are committed to full disclosure of the results of the trial. Findings will be reported in accordance with CONSORT guidelines[32] and we aim to publish in high impact journals. Our patient representatives will assist with dissemination of study results through INVOLVE, other cancer patient groups and organisations including   www.independentcancerpatientsvoice.org.uk.

The funder will take no role in the analysis or interpretation of trial results.

## DISCUSSION

PROSPER will be the largest UK RCT examining the effectiveness of an early, supervised exercise and behavioural support intervention for women at risk of developing shoulder problems after breast cancer surgery. Previous trials in this field have been criticised for being of poor methodological quality and lacking in important outcome measures, such as patient-reported shoulder function and health-related quality of life. Another challenge encountered in previous clinical trials of this population is low participant recruitment, partly due to the short time frame between diagnosis and surgery and perhaps compounded by reluctance to undertake active exercise when faced with a distressing and potentially life-threatening cancer diagnosis. PROSPER aims to recruit 350 patients with newly diagnosed breast cancer to provide empirical data on whether a physiotherapy-led exercise programme is effective for reducing shoulder disability, when delivered in a pragmatic NHS clinical setting. The design and development of this complex intervention was underpinned by multiple stages of work, in line with MRC guidance on the development of complex interventions. A full description of the content of the PROSPER exercise intervention has been submitted elsewhere for publication.

**Acknowledgements** The authors would like to thank all the trial participants. The authors would also like to thank all the physiotherapy staff, surgical oncology teams, breast cancer nurses and research departments collaborating on this study.

**Data Monitoring Committee** Professor Malcolm Reed (Chair), Dr Rhian Gabe, Dr Matthew Maddocks.

**Trial Steering Committee** Professor Steven Duffy (Chair), Dr Anna Kirby, Dr Karen Robb. We dedicate this article to Professor Adele Frances (Deceased) who served on the PROSPER TSC from 2015 to 2016.

**Collaborators** PROSPER Study Group: chief investigator: Professor Julie Bruce. Co-investigators (Grant holders): Professor Sarah E Lamb, Dr Esther Williamson, Dr Ranjit Lall, Professor Stavros Petrou, Mr Alastair M Thompson, Dr John Williams and Dr Catherine Harkin (deceased). Trial co-ordination/administration: Mrs Emma J Withers, Mrs Lauren Betteley, Mr Craig Turner, Mrs Loraine Chowdhury. Senior Project Managers: Mrs Susie Hennings, Mrs Helen Higgins. Research Fellows/Associates: Dr Helen Richmond, Mrs Clare Lait (Physiotherapist), Dr Sophie Rees (Qualitative), Mr Bruno Mazuquin (Research Physiotherapist), Mr Pankaj Mistry (Medical Statistics), Dr Alastair Canaway (Health Economics). Patient representatives: Dr Catherine Harkin (deceased), Mrs Marie van Laar, Mrs Lyn Ankcorn. Surgical leads: Miss Abigail Tomlins, Miss Raghavan Vidya, Ms Pankaj G Roy, Miss Kat McEvoy, Miss Rachel Soulsby. Intervention development: Mrs Clare Lait, Dr Esther Williamson, Dr Cynthia Srikesavan, Mrs Jane Moser, Dr Meredith Newman, Dr Sophie Rees, Mrs Lauren Betteley, Dr Helen Richmond, Dr Beth Fordham, Professor Sarah E Lamb and Professor Julie Bruce. Data programming team: Mr Ade Willis, Mr Henry Adjei. Quality assurance: Ms Claire Daffern.

**Contributors** JB obtained study funding with support from SEL, EW, RL, SP and AMT. JB, SEL, EW, CL, RL, AMT, LB, SR and SP participated in the design of the study. EJW and LB coordinate study administration, acquisition of trial data and administrative support (CT/LC). PM will undertake statistical analysis, under direction of RL, senior trial statistician. AC is responsible for health economic analysis, supported by SP, senior health economist. JB and HR drafted the manuscript. All authors critically revised the manuscript for intellectual content and approved the final manuscript. This trial protocol is published on behalf of the PROSPER Study Group.

**Funding** The PROSPER trial is funded by the National Institute of Health Research Technology Assessment Programme (NiHR HTA), project number 13/84/10. The views expressed in this publication are those of the authors and not necessarily those of the NIHR or Department of Health. This project benefited from facilities funded by Birmingham Science City Translational Medicine Clinical Research and Infrastructure Trials Platform, with support from Advantage West Midlands (AWM). The trial sponsor is the University of Warwick and University Hospitals Coventry and Warwickshire NHS Trust. SL and EW are supported by funding from the National Institute for Health Research (NIHR) Collaboration for Leadership in Applied Health Research and Care Oxford at Oxford Health NHS Foundation Trust, and the NIHR Biomedical Research Unit, Oxford.

**Disclaimer** The views expressed are those of the author(s) and not necessarily those of the NHS, the NIHR or the Department of Health.

**Competing interests** CL provides private physiotherapy to patients with cancer.

**Patient consent** Obtained.

**Ethics approval** Ethical approval was granted from the NHS National Research Ethics Service (NRES) Committee West Midlands (Solihull) (15/WM/0224) on 20 July 2015. Site-specific approvals have been obtained from NHS Research, Development and Innovation departments.

**Provenance and peer review** Not commissioned; externally peer reviewed.

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
