## [Reviewer comments · BMJ Open]

ARTICLE DETAILS

TITLE (PROVISIONAL)	A randomised controlled trial of exercise to prevent shoulder problems in women undergoing breast cancer treatment: study protocol for the Prevention of Shoulder Problems Trial (UK PROSPER Trial)
AUTHORS	Bruce, Julie; Williamson, Esther; Lait, Clare; Richmond, Helen; Betteley, Lauren; Lall, Ranjit; Petrou, Stavros; Rees, Sophie; Withers, Emma; Lamb, Sarah; Thompson, Alastair

VERSION 1 – REVIEW

REVIEWER	An De Groef University of Leuven, Belgium
REVIEW RETURNED	24-Aug-2017

GENERAL COMMENTS	- The focus is on musculoskeletal shoulder problems, to which extent do you take into account neurological problems, e.g. sensory disturbances and neuropathic pain, and lymphoedema in the treatment protocol?- Can you specify what you changed in the protocol based on the pilot study? This may be interesting for other researchers and clinicians to take into account when they deliver an intervention.- In the introduction you elaborate on the studies investigating exercise following breast cancer. Is this the only treatment modality of the intervention? Can you elaborate as well on the evidence for the other modalities?- Rationale for the trial: can you include in this section with which intervention you will compare the 'structured physiotherapy intervention'?- I believe the fact that you test the effectiveness of a 'structured' intervention compared to a 'free' program is important. You may emphasize this in the aims of the methods.- Aim: if I understood well. Aim 1 en 2 are actually for the preparation of the trial. Can you make this more clear? Can you also specify if the pilot study has already finished?- Can you explain more why pts with immediate reconstruction or bilateral surgery are excluded? What is different in the routine postop physiotherapy program for these patients? Wouldn't they benefit from your PROSPER exercise program as well since they are also at high risk?
--

	- Why do you include so many stratification factors? You have a considerable large sample size so randomization should do its work and equally balance the groups? - Control arm: can you clarify if all pts in the control arm will receive the same (2?) brochures? - Is it possible to provide more detailed information on the content, intensity, duration, frequency of the prosper exercise programme? On what is the progression based? In understand this is done pragmatically but can you specify bit more? - To which extent do you take into account physiotherapy delivered by other physical therapists or other treatments for their shoulder problems? - Can you elaborate more on the reliability and validity of the questionnaires? Are they all validated or are they self-developed? Will they be validated? The PASE is for elderly? What with younger participants? -Figure 1: I would suggest to put the outcomes in a second table. Now it is confusing and not clear when assessments are performed, eg. Are all outcomes assessed at baseline (=enrolment), is this before or after surgery? - Table 2: are women with SLNB and mastectomy not at high risk? - Figure 2: can you specify what happens before and after surgery and the duration of the interventions? Note: I am not familiar with economic evaluations so I did not review this part.
--	---

REVIEWER	Ohyun Kwon Yonsei University , South Korea
REVIEW RETURNED	04-Sep-2017

GENERAL COMMENTS	This protocol is well written and organized. However, three issues should be considered.  1. Shoulder conditions, pain history, and shoulder functions (flexibility and strength) should be measured in screening process. Pre- existing shoulder pain and dysfunctions may affect outcome measures. 2. Shoulder exercise comprises ROM exercise, strength exercise, and general physical activity. Exercises should be described in detail for readers. Assessment methods for measuring ROM and strength should be described, ex. Measurement tools et al. And ROM and strength data would be useful information for supporting result of study. 3. Exercise will be performed at home. How authors can monitor compliance of exercise.
---

REVIEWER	Irene de la Rosa Diaz Alcala University, Spain.
REVIEW RETURNED	05-Sep-2017

GENERAL COMMENTS	Functional impairment following breast cancer treatment is an important target for physiotherapists nowadays, so I find the paper interesting and pragmatic. Relating to methods, sample homogeneity is needed to avoid confusing factors, thus contralateral breast cancer or relapses should be excluded. Blinding techniques should be clarified. The PROSPER approach should be well-detailed as an exercise-performance explanation is needed to enable study repetition. Likewise, number and duration of the physiotherapy sessions should be mentioned. About sample size calculation, pilot data is for calculating the sample size of the clinical trial. However, authors used data from other study instead. That issue should be justified. Concerning to outcomes, the assessment method of secondary variables and the instruments used should be detailed. In addition, the election of DASH questionnaire has to be justified since there are other more specific questionnaires to assess shoulder functions (i.e. SPADI or OSS). Relating to discussion section, it should be more explained and conclusions have to be added. Dr. Irene de la Rosa Diaz.
--

VERSION 1 – AUTHOR RESPONSE

Reviewer 1

1. The focus is on musculoskeletal shoulder problems, to which extent do you take into account neurological problems, e.g. sensory disturbances and neuropathic pain, and lymphoedema in the treatment protocol?

Response: Women randomised to the treatment arm are reviewed by the PROSPER physiotherapist during each face to face appointment and are assessed for signs of lymphoedema and painful symptoms. The PROSPER treatment protocol includes recommended pathways to follow based upon clinical findings e.g. consider referral to lymphoedema nurse if painful arm swelling or heaviness; refer to GP if pain intensity scores greater than 7/10 on visual analogue scale. We have limited the description of the exercise intervention in the trial protocol paper as we have written a separate detailed paper describing intervention development and content. No changes have been made to the manuscript.

2. Can you specify what you changed in the protocol based on the pilot study? This may be interesting for other researchers and clinicians to take into account when they deliver an intervention.

Response: These were mostly minor changes to the wording on trial-related and patient-facing documents after feedback from qualitative interviews with newly-diagnosed breast cancer patients and breast cancer survivors. Overall, patients and recruitment staff were positive about the proposed study. We have edited the paragraph 'Internal Pilot Study' (page 11) by adding "Changes were made to patient-facing materials and to exercise intervention materials. Easy to use pocket-sized laminated cards with details of the PROSPER trial inclusion/exclusion criteria and shoulder screening criteria were produced for recruitment staff. Additional telephone or face-to-face appointments were added to the exercise intervention to allow for flexibility during ongoing cancer treatment".

3. In the introduction you elaborate on the studies investigating exercise following breast cancer. Is this the only treatment modality of the intervention? Can you elaborate as well on the evidence for the other modalities?

Response: This is a good point and we have elaborated on this in the trial intervention paper which is currently under review with BMC Health Services Research. In the intervention manuscript, we describe the evidence-base for behavioural strategies, including NICE guidance for behavioural change – thus, how best to promote adherence to self-management interventions. The PROSPER intervention also implemented strategies from the NHS Health Trainer Manual, developed by behaviour change experts. We have not elaborated on this within the trial protocol paper which focuses on the rationale for the trial, aims, methods and study design issues etc.

4. Rationale for the trial: can you include in this section with which intervention you will compare the 'structured physiotherapy intervention'?

Response: Thank you for highlighting this. 'Compared to usual care' has been added to the first sentence under the 'Rationale for a trial' section, on page 6.

5. I believe the fact that you test the effectiveness of a 'structured' intervention compared to a 'free' program is important. You may emphasize this in the aims of the methods.

Response: The term 'structured' has been added to the Abstract (line 4) and under the second aim, within the Methods section on page 6.

6. Aim: if I understood well. Aim 1 and 2 are actually for the preparation of the trial. Can you make this more clear? Can you also specify if the pilot study has already finished?

Response: This is correct, the first two aims were to develop an exercise intervention and to establish whether the trial was acceptable to patients and feasible to deliver. The third aim was then to undertake a definitive RCT to examine the clinical and cost-effectiveness of an early supervised exercise programme compared to usual care for women at high risk of shoulder problems after treatment for breast cancer. These aims are published on the NIHR HTA website as the study was funded on this basis. Rather than restrict our description to the main trial only, we have been explicit and transparent by referring to all aims in this protocol paper. The pilot study was completed as planned; this has been clarified within the paragraph on page 11 – 'Internal pilot study'. The sentence 'The pilot study was completed as planned, and the funder approved progression to full trial' has been added.

7. Can you explain more why patients with immediate reconstruction or bilateral surgery are excluded? What is different in the routine postop physiotherapy program for these patients? Wouldn't they benefit from your PROSPER exercise program as well since they are also at high risk?

Response: We very much agree that women undergoing reconstruction and/or bilateral breast surgery are at high risk of developing shoulder problems and could potentially benefit from an exercise programme. However it would be challenging to develop a 'one-size fits all' structured exercise intervention for non-reconstructive surgery and the many different reconstructive surgery procedures, which include implants or autologous tissue flaps, including abdominal (e.g. DIEM, SIEA TRAM), buttock and thigh flaps (e.g. GAP, TUG, PAP) and/or back muscle flaps (LAT). These would each require careful consideration of movement restrictions in the acute postoperative period. Reconstructive breast surgery is classed as major surgery whereby patients are hospitalised for 7-10 days postoperatively; more drains are left in situ and these women are at greater risk of postoperative complications, including wound infection.

These are a different clinical group to those admitted as day cases for mastectomy/WLE. The current UK model is for non-reconstructive breast surgery is discharge within a day, as per the 23hour Ambulatory Surgery model of care. The PROSPER intervention was designed specifically for these patients. It would be overly complex to design and test two or more exercise interventions for different

clinical groups within the same trial framework. We do believe there is potential for a separate trial investigating the effectiveness of postoperative exercise for women undergoing reconstructive breast surgery.

Re. bilateral non-reconstructive surgery, this was a pragmatic decision. It is challenging to design data collection tools to capture outcomes from women having surgery on both breasts and both axilla. Our data collection tools were designed to record painful symptoms, swelling, functional problems etc. on the operated side only. This was based on our experience of previous studies investigating recovery after breast cancer surgery.

8. Why do you include so many stratification factors? You have a considerable large sample size so randomization should do its work and equally balance the groups?

Response: It is usual practice to stratify by site to account for potential variation in clinical care. We stratified by timing of entry to the study as a small proportion of women are eligible for the trial postoperatively, if referred for radiotherapy to the axilla. This decision is made after pathology reports are obtained. Also a small number of women will undergo repeat surgery, thus readmission for excision of surgical margins. Stratification was undertaken to ensure there was equal distribution across treatment arms for these treatment subgroups.

9. Control arm: can you clarify if all pts in the control arm will receive the same (2?) brochures?

Response: Yes, all participants receive the two information leaflets from Breast Cancer Care. This has been clarified on page 9 by adding 'All participants' in line 1 and 'These two information leaflets were given to all patients....' on line 8 of that paragraph.

10. Is it possible to provide more detailed information on the content, intensity, duration, frequency of the prosper exercise programme? On what is the progression based? In understand this is done pragmatically but can you specify bit more?

Response: Apologies to give the same response as before, but we have restricted the description of the exercise intervention in this protocol paper to the essential facts. We acknowledge that the reviewer is an experienced physiotherapy researcher and is understandably looking for more detailed information about frequency, duration, intensity, progression etc. We can certainly provide more information within this manuscript however seek guidance from the Editorial team on this matter. We have a separate intervention publication as per recommended guidance from the EQUATOR/CONSORT network – the intervention paper adheres to the Template for Intervention Description and Replication (TiDIER) framework.

11. To which extent do you take into account physiotherapy delivered by other physical therapists or other treatments for their shoulder problems?

Response: We ask all trial participants to report any attendance with any other healthcare professional – these data will also be used for health economic analysis. Participants complete questionnaires at six weeks, six months and 12 months, and report any contacts with GPs, physiotherapists and any other healthcare professionals. As the trial is pragmatic, we do not restrict contact and any patient can attend a non-PROSPER physical therapist at any time.

12. Can you elaborate more on the reliability and validity of the questionnaires? Are they all validated or are they self-developed? Will they be validated? The PASE is for elderly? What with younger participants?

Response: All of the questionnaires have been validated. We carefully considered a number of questionnaires to assess shoulder function. There is reasonable evidence of validity and reliability for the DASH scale. A review summarised the psychometric properties of nine commonly used instruments designed to measure symptoms and function of the shoulder (Angst et al, 2011). In summary, the DASH questionnaire was reported to be the most widely used and thoroughly tested instrument. This scale captures symptoms and function of the upper limb rather than the shoulder joint per se. This is important because there is good evidence to show that women experience problems with the upper limb after breast cancer treatment, not just the shoulder joint. Although surgery and radiotherapy target the breast and axilla area, treatment side effects can impact upon the hand, arm and shoulder e.g. potential trauma to the nervous and lymphatic systems, leading to arm swelling, problems with grip strength etc.

Re the other measures, we have experience of capturing pain outcomes after breast cancer treatment. The DN4 Neuropathic pain scale has been used in numerous population-based and surgical cohort studies, including breast cancer surgery. It is short, easy to complete and has reasonable correlation with objective tests of nerve dysfunction (hyperalgesia etc).

Re inclusion of items from the Physical Activity Scale for the Elderly (PASE). One component of the exercise intervention is to encourage physical activity throughout the recovery period. Again, we considered a number of different physical activity measures – many of these questionnaires were too detailed and lengthy. We wanted a brief indicator of walking and physical activity to compare change from recruitment to follow-up. The PASE was originally developed for use in epidemiological surveys with people aged 65 years and older. Given that the mean age of our PROSPER pilot sample was 61 years, and that our PPI group felt that the PASE questions were easy and straightforward to answer, we opted to use items from this questionnaire.

13. Figure 1: I would suggest to put the outcomes in a second table. Now it is confusing and not clear when assessments are performed, eg. Are all outcomes assessed at baseline (=enrolment), is this before or after surgery?

Response: We used the template figure for trial protocol papers as per SPIRIT 2013. However we agree with the reviewer that this is not a very user-friendly figure! We have modified it from the recommended template – it has been split into Figure 1 and Table 3. This should be clearer for readers.

14. Table 2: are women with SLNB and mastectomy not at high risk?

Response: No, women having more extensive axillary surgery are at greater risk of shoulder problems than those having sentinel lymph node biopsy. Type of breast surgery is not a major independent factor for adverse postoperative outcomes of shoulder dysfunction or chronic postoperative pain. There is recent evidence demonstrating that other factors, including axillary clearance, are associated with onset and persistent of chronic pain e.g Meretoja et al (2017, J Clin Oncol).

15. Figure 2: can you specify what happens before and after surgery and the duration of the interventions?

Response: Patients are recruited and randomised before surgery. The first treatment appointment is booked for 7 to 10 days postoperatively, as described on page 9. The second appointment is booked for between four to six weeks postoperatively (page 10, line 4). The third appointment is recommended for between 12 and 16 weeks postoperatively (page 10, line 7). The pathway differs slightly for women with postoperative entry to the trial, but the exercise programme should commence

as soon as possible, within six weeks of surgery for this latter group (as described on page 10, lines 8-10). This again described more fully in the intervention paper.

Note: I am not familiar with economic evaluations so I did not review this part.

Response: Thank you for your comments.

Reviewer: 2

This protocol is well written and organized. However, three issues should be considered.

1. Shoulder conditions, pain history, and shoulder functions (flexibility and strength) should be measured in screening process. Pre-existing shoulder pain and dysfunctions may affect outcome measures.

Response: Pain intensity, pain character, and shoulder function are captured in baseline questionnaires which are completed by all recruited participants. Further assessment is undertaken by trained physiotherapists on women randomised to the exercise arm. We agree that pain history and dysfunction may affect outcome measures therefore outcomes will be adjusted for baseline DASH scores – we aim to present unadjusted and adjusted values, as per the full Statistical Analysis Plan (SAP).

2. Shoulder exercise comprises ROM exercise, strength exercise, and general physical activity. Exercises should be described in detail for readers. Assessment methods for measuring ROM and strength should be described, ex. Measurement tools et al. And ROM and strength data would be useful information for supporting result of study.

Response: These have been described in the intervention development paper. We can send a copy of this manuscript for the Editorial team if this is required.

3. Exercise will be performed at home. How authors can monitor compliance of exercise.

Response: All participants are given a personalised folder which includes a menu of exercises and diaries to record information about completed exercises - where and when they were done. Women are provided with stamped addressed envelopes to return diaries to the main study office every month. A six month supply of diaries is given to everyone randomised to the intervention. We fully acknowledge the limitations of self-report but the use of diaries is standard practice for home-based exercise interventions.

Reviewer: 3

Functional impairment following breast cancer treatment is an important target for physiotherapists nowadays, so I find the paper interesting and pragmatic. Thank you.

1. Relating to methods, sample homogeneity is needed to avoid confusing factors, thus contralateral breast cancer or relapses should be excluded. Blinding techniques should be clarified.

Response: Patients with known contralateral breast cancer at time of recruitment were ineligible. Women with cancer recurrence remained in the trial if detected after randomisation. This is impossible to predict therefore a number of women will undergo further surgery for cancer relapse during the course of 12 month follow-up post-randomisation. We will collect clinical information from the medical records of all participants reaching the 12 month follow-up time point.

Re blinding, this was described on page 8. This has been edited slightly to read as follows: 'However, receipt and handling of outcome data collection is blinded, thus data entry of returned postal questionnaires, data cleaning and interim statistical analyses are conducted without knowledge of treatment allocation (blinded).'

2. The PROSPER approach should be well-detailed as an exercise-performance explanation is needed to enable study repetition. Likewise, number and duration of the physiotherapy sessions should be mentioned.

Response: We fully agree with the reviewer that a thorough description of exercises is required. We found this a challenge when designing the PROSPER intervention, in that only a small number of published clinical trials actually describe what was delivered. Interventions were often reported as 'stretching and strengthening' but no information was given on the actual movements or whether resistance bands or weights were used. These are all fully described in the trial intervention paper. On page 9, we do describe the number of sessions with physiotherapists – a minimum of three face-to-face and a maximum of six sessions. We have edited the next sentence to read: 'A more detailed description of intervention development and final content has been submitted elsewhere for publication.'

3. About sample size calculation, pilot data is for calculating the sample size of the clinical trial. However, authors used data from other study instead. That issue should be justified.

Response: The pilot study was not undertaken to determine the sample size, it was conducted to assess feasibility and acceptability. The sample size calculation was indeed based on a previous clinical trial (Beurkensens, 2007). We considered the expected effect size for the DASH based on the between group difference found in the Dutch trial testing a similar intervention in a breast cancer population, albeit with a smaller sample size (30 patients). This RCT found a between group difference of 7 points on the DASH at 6 months (SMD 0.36). We have moved the sample size calculation paragraph before the paragraph describing the internal pilot study, to aid clarity (page 12).

4. Concerning to outcomes, the assessment method of secondary variables and the instruments used should be detailed. In addition, the election of DASH questionnaire has to be justified since there are other more specific questionnaires to assess shoulder functions (i.e. SPADI or OSS).

Response: We have added a few sentences to the Outcomes paragraph under Methods section on page 10, as follows: 'We considered other patient-reported outcome measures, including shoulder-specific scales, however selected the DASH because it captures symptoms and function of the upper limb rather than the shoulder joint per se.'

There is good evidence to suggest that women experience a variety of difficulties and restrictions after breast cancer treatment, affecting the hand, arm and shoulder. Functional impairment to the arm can affect performance of simple daily activities, including writing, opening or closing jars, lifting and/or holding shopping bags.'

The secondary outcomes paragraph has been edited. It now reads as follows:

'Secondary outcomes include health-related quality of life (EuroQol EQ-5D-5L and Short-Form-12), DASH sub-scores, and surgical adverse events including pain (acute, chronic, neuropathic pain) surgical site infection and lymphoedema. A numerical rating scale (NRS) 0-10 and Doleur Neuropathique Questionnaire (DN4) are used to collect pain intensity and pain character. The Functional Assessment of Cancer Therapy-Breast (FACT-B4) subscale captures arm tenderness, numbness, painful movement and stiffness. We added items to capture arm heaviness and swelling

as self-report indicators of lymphoedema. Data on exercise/mobility are collected to allow comparisons in physical activity (selected items from the Physical Activity Scale for the Elderly (PASE)). Healthcare resource use is recorded for economic analyses.

5. Relating to discussion section, it should be more explained and conclusions have to be added.

Response: We have followed the format of previous trial protocol papers and not written a lengthy discussion or conclusion. We can discuss the trial findings, strengths and limitations of the study within the main RCT publication and HTA monograph. However, the following sentence has been added to reiterate that a manuscript of the exercise intervention will be published separately. 'A full description of the PROSPER exercise intervention has been submitted elsewhere for publication.'(page 18). Currently, we have not expanded the Discussion section any further but can do this with permission from the Editorial team.

A number of comments from the reviewers relate to the trial physiotherapy intervention however we are awaiting feedback from the Editorial team at BMC Health Services Research (submitted April 2017).

Thank you again for consideration of our manuscript. I can confirm that all authors have read and approved these edits.

VERSION 2 – REVIEW

REVIEWER	An De Groef University of Leuven, Belgium Department of Rehabilitation Sciences
REVIEW RETURNED	15-Nov-2017
GENERAL COMMENTS	All my comments were carefully addressed and clarified so I have no further comments and suggest to accept the paper.
REVIEWER	Ohyun Kwon Yonsei University, South Korea
REVIEW RETURNED	14-Nov-2017
GENERAL COMMENTS	Authors may need to confirm whether the Physical Activity Scale for the Elderly is appropriate to measure the level of exercise/mobility in this study. - Authors well revised that I recommended. - Authors may need to confirm whether the Physical Activity Scale for the Elderly is appropriate to measure the level of exercise/mobility in this study.
REVIEWER	Irene de la Rosa Díaz Alcala University. Madrid. Spain.
REVIEW RETURNED	28-Nov-2017
GENERAL COMMENTS	Overall, corrections were included properly. However, the detailed intervention of study, the PROSPER exercise programme, remains lacking. The following sentence "A more detailed description of

	intervention development and final content has been submitted for publication" indicates the intervention of study is explained but not within this paper. That is needed to guarantee the reproducibility of the study. Moreover, discussion and limitations sections remained poor, methodology and drawbacks must be discussed. Justification of DASH questionnaire election is indicated but a reference must be cited.
--	---

VERSION 2 – AUTHOR RESPONSE

Reviewer 1 (de Groef)

All comments carefully addressed. Thank you.

Reviewer 2 (Kwon)

Please confirm whether the PASE scale is appropriate for this study.

The PASE section on page 12, paragraph 1, has been edited and a reference added. We have added the following: "The PASE was designed for use with older adults but has been validated for use in clinical trials recruiting patients aged 55 years and older." (Reference 17 - Washburn et al. The physical activity scale for the elderly: evidence for validity).

Response: Please note there are also epidemiological studies and clinical trials whereby the PASE scale has been used to measure activity in cancer patients undergoing active treatment and studies of cancer survivors. Various studies have recruited middle-aged patients rather than older adults per se. We have not added these references as do not want to extend the length of the article. We are happy with the face validity of the PASE items. Finally, the PASE questions are included to give a crude-level indicator of overall self-reported physical activity, however physical activity is not the primary outcome for the trial.

Reviewer 3 (de la Rosa Diaz)

a) The reviewer has asked again for more information about the trial intervention. The description of the exercise intervention now spans two pages (from pages 9 to 11). We have added in some background information about the Cochrane review that was considered when developing the intervention and that we consulted experts in the field of cancer rehabilitation and health psychology. The duration, timing and assessment is described, as is the flexible nature of the intervention. The types of movements have been added e.g. active range of movement, active-assisted ROM, stretches and strengthening. We do hope this is enough detail for the Editor. Furthermore, we have uploaded a copy of the intervention description manuscript submitted to BMC Health Services Research as evidence of submission (decision pending since April 2017).

b) Please include limitations in the Discussion section.

Response: We have added another few sentences to the Discussion. We followed usual format for our previous protocol papers published by the BMJ Open (e.g. Prevention of Falls Injury trial protocol, WOLFF trial protocol and SCOTS bariatric study protocol), whereby only a brief discussion was included as it is rather early to consider all the methodological shortcomings. We have designed the trial to the best of our ability and will discuss limitations and/or challenges when the trial is completed and reported.

Thank you again for your help. We look forward to hearing from you.

J Bruce, on behalf of the PROSPER Study Group